# An Efficient and Rapid Protocol for Somatic Shoot Organogenesis from Juvenile Hypocotyl-Derived Callus of Castor Bean cv. Zanzibar Green

**DOI:** 10.3390/biotech13030025

**Published:** 2024-07-04

**Authors:** Danaya V. Demidenko, Nataliya V. Varlamova, Taisiya M. Soboleva, Aleksandra V. Shitikova, Marat R. Khaliluev

**Affiliations:** 1All-Russia Research Institute of Agricultural Biotechnology, Timiryazevskaya 42, 127434 Moscow, Russia; frankenvini1998@mail.ru (D.V.D.); nv_varlamova@rambler.ru (N.V.V.); 2Agrobiotechnology Institute, Russian State Agrarian University—Moscow Timiryazev Agricultural Academy, Timiryazevskaya 49, 127434 Moscow, Russia; soboleva.taisia@yandex.ru (T.M.S.); plant@rgau-msha.ru (A.V.S.)

**Keywords:** efficiency of indirect in vitro shoot organogenesis, explant source, morphological characteristics of callus tissue, plant growth regulators (PGRs), plantlets adaptation to ex vitro conditions, *Ricinus communis* L., shoot rhizogenesis

## Abstract

Aseptic seedlings of different ages derived from surface-sterilized mature seeds were applied as an explant source. Various explants such as 7- and 21-day-old hypocotyl fragments, 42-day-old nodal stem segments, and transverse nodal segments of stem, as well as leaf petioles, were cultured on the agar-solidified Murashige and Skoog (MS) basal medium supplemented with 0.1 mg/L IAA, 5 mg/L AgNO_3_ and different types and concentrations of cytokinin (1 mg/L zeatin, 0.25 mg/L thidiazuron (TDZ), and 5 mg/L 6-benzylaminopurine (6-BAP)). Consequently, it was found that 7- and 21-day-old hypocotyl fragments, as well as nodal stem segments obtained from adult aseptic seedlings, are characterized by a high explant viability and callus formation capacity with a frequency of 79.7–100%. However, the success of in vitro somatic shoot organogenesis was significantly determined not only by the culture medium composition and explant type but also depending on its age, as well as on the size and explant preparation in cases of hypocotyl and age-matched nodal stem fragments, respectively. Multiple somatic shoot organogenesis (5.7 regenerants per explant) with a frequency of 67.5% was achieved during 3 subcultures of juvenile hypocotyl-derived callus tissue on MS culture medium containing 0.25 mg/L TDZ as cytokinin source. Castor bean regenerants were excised from the callus and successfully rooted on ½ MS basal medium without exogenous auxin (81%). In vitro plantlets with well-developed roots were adapted to ex vitro conditions with a frequency of 90%.

## 1. Introduction

Castor bean (*Ricinus communis* L.) is the most valuable non-food crop among members of the Euphorbiaceae family. According to the Food and Agriculture Organization of the United Nations (FAO), the world’s total castor bean seed production in 2022 was approximately 1.8 million tons, cultivated on an area of 1.2 million hectares [1]. Castor bean, as a heat-loving crop, is predominantly cultivated in tropical, subtropical, and temperate regions. India, China, and Brazil are the world’s leaders in the commercial cultivation of castor bean seeds, accounting for more than 90% of the total harvested area [2]. In 2022, castor bean as an oilseed crop was grown in Russia’s southern zone (Krasnodar and Stavropol territories, as well as the Rostov region); however, its total harvested area was scanty and amounted to only 107 hectares [1]. However, castor bean was considered to be a strategic value for the USSR in the mid-twentieth century, and its production in 1940 occupied 172 thousand hectares [3]. Thus, Russia is currently highly import-dependent on castor bean seeds for oil production.

Regardless of purification degree, both cold-pressed and hot-pressed castor oils are widely used in various manufacturing, including aerospace, textile, paper, and perfume industries, in plastic production, and as a laxative in traditional medicine [4,5]. Due to the high content of ricinoleic acid triglyceride (89–92%), the main component of castor oil, it is commonly applied in the cosmetic industry, providing analgesic, bactericidal, anti-inflammatory, and regenerative properties [6,7]. Additionally, castor bean, as a high-oil industrial crop (oil content ~55% of dry seed), is considered a promising feedstock alternative for biodiesel production [8,9].

Many by-products are left behind after oil seed pressing, which is also of great value as highly nutritious feed additives for livestock [10,11] and fish [12]. However, castor seed meal and castor press cake currently cannot be used in industrial recycling due to the presence of two highly toxic compounds: toxalbumin called ricin, which accumulates predominantly in the seed coat and promotes a direct cytotoxic effect by inhibition of protein biosynthesis through enzymatic ribosome inactivation; and ricinine, a small alkaloid, present in the castor seeds, the toxicity of which is significantly less than ricin [13,14]. 

The development of highly effective methods for ricin and ricinine decomposition in castor meal and castor press cake may solve this problem; however, traditional thermal-detoxified or chemical-detoxified treatments have proven to be unsuitable, resulting in the degradation of protein and nutrients [15]. In this regard, the utilization of alternative biological detoxification approaches is very promising, for example, applying genetic engineering strategies for producing castor bean plants with inhibited or reduced ricin biosynthesis through RNA interference or CRISPR/Cas genome editing [16,17].

Regardless of the genetic transformation techniques for producing transgenic or genome-editing castor bean plants, the crucial prerequisite is to develop an effective protocol of in vitro full-fledged multiple shoot induction from various explants. A literature review shows that shoot regeneration is achieved through both direct [18,19,20] and indirect [21,22,23] somatic organogenesis from isolated protoplasts [24], juvenile hypocotyl fragments with shoot apical meristem [19,20,25], fragments of hypocotyl [21,23] and epicotyl [18], cotyledon petioles [23], cotyledons [22,26], and immature [27] and mature [16,28] embryos.

Primary explants of castor bean with an apical shoot meristem are most suitable for clonal micropropagation [19,29] but not for direct methods or *Agrobacterium*-mediated genetic transformation due to the high probability of regeneration of chimeric plants during direct organogenesis since they are formed not from one cell, but from a group of cells [30]. On the contrary, the developed protocols for indirect somatic shoot organogenesis from explants without an apical meristem are characterized by less efficiency and significantly depend on the genotype [21]. Thus, the highest frequency of indirect shoot organogenesis from hypocotyl explants obtained from 10–12-day-old castor bean seedlings of the GC-3 and JP-65 lines was 72.3% and 77.7%, respectively, when cultured on the agar-solidified Murashige and Skoog (MS) basal medium [31] supplemented with 0.5 mg/L kinetin and 0.25 mg/L 6-benzylaminopurine (6-BAP). At the same time, other explants used in the experiment (cotyledons, epicotyls, young true leaves, and mature embryos) were characterized by an extremely low callus formation ability [21]. Ganesh Kumari et al. (2008) have reported that the callus induction frequency from cotyledon explants of castor bean cv. TMV 6 varied from 5.2 to 69.5% depending on the types and concentrations of plant growth regulators (PGRs) in MS culture medium [22]. 

Our previous studies established high callus formation ability from hypocotyl fragments and cotyledon petioles of the castor bean cvs. Zanzibar Green and Impala Bronzovaya with a frequency of 60 to 100% [23]. The highest frequency of somatic shoot organogenesis (19.3 and 43.3% for the cvs. Zanzibar Green and Impala Bronze, respectively) was detected when cotyledon petioles were cultured on MS medium supplemented with 2 mg/L zeatin in combination with 0.1 mg/L indole-3-acetic acid (IAA). When cultured hypocotyl fragments were isolated from adult aseptic seedlings, indirect organogenesis of single shoots was observed with a frequency of 10% in both castor bean genotypes [23]. Accordingly, the low regeneration capacity from studied callus-derived castor bean explants of cv. Zanzibar Green did not allow us to carry out genetic transformation. Thus, the presented article is devoted to optimizing the explant source and culture conditions for the development of an effective protocol of in vitro castor bean shoot organogenesis cv. Zanzibar Green.

## 2. Materials and Methods

### 2.1. Plant Material

The castor bean seeds (*R. communis* L., 2n = 2x = 20) of the cv. Zanzibar Green (Gavrish, Russia) were used as initial plant material.

### 2.2. Obtaining Aseptic Donor Seedlings. Culture Conditions

In vitro aseptic donor castor bean seedlings were produced by seed sterilization in 96% ethanol for 20 min and in 50% water solution (*v*/*v*) of a fresh commercial bleach Belizna (<5% NaOCl) (Spektr, Saint Petersburg, Russia) for 40 min. Surface-sterilized seeds were washed with distilled water three times for 10, 15, and 20 min each, respectively, and then a hard seed coat was mechanically removed. Scarified castor bean seeds germinated in culture vessels containing 0.8% agar-solidified (*w*/*v*) MS medium with 3% sucrose. The pH was adjusted to 5.7–5.8 before autoclaving at 121 °C for 20 min. The cultures were maintained in a climate chamber WLR-351H (Sanyo, Tokyo, Japan) under 25/23 (day/night) ±1 °C, with fluorescence light (80 µmol m^−2^s^−1^) during a long-day photoperiod (16 h light/8 h dark).

### 2.3. In Vitro Callus Induction and Somatic Shoot Organogenesis

Induction of indirect shoot organogenesis of castor bean was carried out from various explants derived from aseptic seedlings of different ages: (1) 7-day-old hypocotyl fragments (0.5–1.0 cm long); (2) 21-day-old hypocotyl fragments (1.0–1.5 cm long); (3) 42-day-old nodal segments of stem (0.5–1.0 cm long); (4) 42-day-old transverse nodal segments of stem (2–3 mm thickness); and (5) petioles of first true leaves (1.0–1.5 cm long). To obtain juvenile hypocotyl fragments, sterilized seeds were cultured for 7 days, after which the germinating seedlings were removed with a scalpel, and the middle part of the explant was cut off. After 7 days of pre-culture of explants on the callus induction medium, a small part from the hypocotyl edges was additionally cut off to exclude the presence of root and apical meristem.

Initial explants were cultured on MS basal culture medium supplemented with 0.1 mg/L IAA (Phyto Tech. Lab, Lenexa, KS, USA) as an auxin source, 5 mg/L AgNO_3_ (Phyto Tech. Lab, Lenexa, KS, USA) as an inhibitor of ethylene biosynthesis, and various types and concentrations of cytokinin: 1 mg/L zeatin (Phyto Tech. Lab, Lenexa, KS, USA) (MS1); 0.25 mg/L thidiazuron (TDZ) (Phyto Tech. Lab, Lenexa, KS, USA) (MS2); and 5 mg/L 6-BAP (Phyto Tech. Lab, Lenexa, KS, USA) (MS3). PGRs were dissolved in distilled water, filter-sterilized (0.22 μm Millipore, Burlington, MA, USA), and added to the cooled culture medium after autoclaving. Various explants were subcultured onto the fresh culture medium every 14 days.

### 2.4. In Vitro Root Induction of Regenerants and Plantlets Adaptation to Soil Conditions

When shoots regenerated from 7-day-old hypocotyl fragments were about 1.0–1.5 cm long, they were excised from callus and transferred onto root induction medium (RIM) containing half-strength agar-solidified MS medium (½ MS) without PGRs or supplemented with 1 mg/L indole-3-butyric acid (IBA). Regenerants with well-developed roots were acclimatized to ex vitro conditions. Subsequently, adapted plants were grown in plastic pots filled with a sterilized mixture of garden soil and sand (3:1) under a greenhouse (23–25/19–20 °C (day/night) time temperature, humidity 60–70%, illumination 2500 lx). 

### 2.5. Estimation of the Efficiency of an In Vitro Morphogenetic Responses

The site of callus formation was determined when explants were cultured on MS medium with different PGR contents. Additionally, callus tissue was also characterized by morphological characteristics such as color and consistency. Explant viability was calculated after 21 days of culture as the percentage number of viable explants from the total number. The efficiency of morphogenetic responses was assessed after 42 days of in vitro culture using such values as the frequencies of callus formation and somatic shoot organogenesis, as well as the average number of regenerated shoots per explant. The frequency of callus formation, expressed as a percentage, was determined as the ratio between the number of explants that produced callus tissue and the total cultured explants. The frequency of somatic shoot organogenesis, expressed as a percentage, was calculated as the ratio of the explant number in which shoot regeneration occurred to the total explant number with callus induction. The average shoot number per explant was determined as the ratio of the number of regenerants to the total number of explants in which indirect shoot organogenesis occurred. The frequency of rhizogenesis, expressed as a percentage, was estimated as the ratio of rooted regenerants to the total number. 

### 2.6. Statistical Processing of Experimental Data

Each experimental variant was performed in three biological replicates. The number of samples in each biological replicate varied depending on the explant type (n ≥ 10).

Statistical treatments of experimental data were performed at a 5% significance level (α = 0.05) using the two-way analysis of variance (ANOVA) and Duncan’s multiple range tests with AGROS software (version 2.11 1993–2000, Moscow, Russia), as well as standard MS Excel 2019 software packages. The percentage values such as explant viability, frequencies of callus induction, shoot organogenesis, and root formation were arcsin$\sqrt{X}$ transformed prior to the ANOVA test. The average number of shoots per explant were $\sqrt{X+1}$ transformed prior statistical analyses.

## 3. Results

### 3.1. Influence of Culture Medium Composition on Explant Viability

During the first and second subcultures, the number of necrotic explants was recorded. Consequently, the explant viability was assessed after 21 days of culture on MS medium with different PGRs (Table 1). Hypocotyl explants varying in age, 42-day transverse stem segments, and nodal stem fragments were characterized by high viability (91.7–100%) for all treatments. Stem segments cultured on MS3 medium supplemented with 6-BAP (75.0%). The viability of leaf petioles was significantly lower (63.3–83.3%) compared to other explants when cultured on MS1–MS3 induction media. 

### 3.2. Effects of Explant Source and Culture Medium Composition on In Vitro Callus Induction

#### 3.2.1. Morphological Characteristics of Callus Tissue

During the first and second subcultures, a primary callus formed on the initial various explants was identified and characterized by color and consistency. Additionally, the site of callus formation and morphogenetic response were observed (Table 2 and Figure 1). 

Hypocotyl-derived primary callus tissue produced from juvenile and adult castor bean seedlings differed significantly on the above-mentioned morphological characteristics (Figure 1a–c). The same was observed for nodal stem segments, depending on their size and the explant preparation, as well as culture medium composition (Figure 1e–g). Thus, a morphogenic dense callus of green color was formed on the cut from the shoot meristem side of 7-day-old hypocotyl fragments (Figure 1a,b). Brown and friable (MS1 and MS3 media), as well as light green and dense (MS2 medium) calli formed on the cut from the root meristem side of 7-day-old hypocotyl fragments, were non-morphogenic. In contrast to juvenile hypocotyl explants, the primary callus produced at both ends of the cut from 21-day-old hypocotyl fragments was characterized as friable, brown or yellow-brown, and non-morphogenic (Figure 1c,d). Morphogenic dense calli of yellow or green color, formed at both ends of the nodal stem cut, were achieved on MS1 and MS3 culture media. On the contrary, transverse nodal segments of the stem are characterized by the formation of yellow-brown or green dense and morphogenic callus tissue on all culture media, which occupied the adaxial and abaxial surfaces of the whole explant (Figure 1e,f). Morphogenic dense callus of light green color on leaf petioles was obtained only when cultured on MS1 medium. At the same time, shoot organogenesis from petioles-derived primary callus was produced only on the cut from the stem side. 

During the third subculture, on the surface of the primary callus derived from various explants, the formation of secondary friable callus cultures of white or brown color occurred. In all treatments, secondary calli were non-morphogenic (Figure 1c,d,g,i).

#### 3.2.2. Callus Formation Efficiency

The results of the two-way ANOVA test showed statistical differences at a 5% significance level in callus formation frequency between different castor bean explants and the studied culture media. However, no significant difference was observed between the interaction of factors “culture medium” × “explant” (Table 3). 

Juvenile fragments of hypocotyl were characterized by the maximum ability to callus formation when cultured on MS1–MS3 media (Figure 2). On average, for all studied media variants, no statistical differences were revealed between 7-day-old and 21-day-old hypocotyl fragments (99.0%), as well as 42-day-old nodal fragments of stem (99.2%) in the callus formation frequency. According to the callus formation ability, other types of explants were arranged in the following sequence: 7-day-old hypocotyl fragments (100%) > 42-day-old transverse nodal fragments of stem (92.4%) > leaf petioles (79.7%).

The MS1 culture medium, containing zeatin as a cytokinin source, provided a significantly higher frequency of callus formation from 42-day transverse nodal segments of stem and leaf petioles compared to MS3, supplemented with 5 mg/L 6-BAP. However, on average, for all types of explants, no statistically significant differences were found between MS1, MS2, and MS3 culture media in the callus formation frequency (97.5, 97.0, and 87.7%, respectively). 

### 3.3. Effects of Explant Source and Culture Medium Composition on In Vitro Induction of Somatic Shoot Organogenesis

The efficiency of somatic shoot organogenesis significantly depends on the explant type, as well as culture medium composition (Figure 3a). Additionally, radically age-dependent differences in the morphogenetic response of hypocotyl segments were observed. The frequency of indirect somatic shoot organogenesis from juvenile hypocotyl segments of castor bean varied from 33.2 (MS3) to 67.5% (MS1), depending on the culture medium treatments. Additionally, zeatin (MS1) and TDZ (MS2) in the culture media (Figure 4a,b) induced a greater stimulating effect on in vitro shoot organogenesis from 7-day-old hypocotyl segments than 6-BAP (MS3) (Figure 4c). On the contrary, callus tissue formed on hypocotyl explants obtained from adult aseptic seedlings was non-morphogenic. Regardless of the PGR composition in the culture medium, somatic organogenesis of shoots from 21-day-old hypocotyl segments did not occur.

In the case of age-matched nodal stem explants, in vitro shoot organogenesis capacity was significantly determined by its size and the explant preparation. Thus, the frequency of somatic shoot organogenesis during the culture of nodal segments of stem 0.5–1.0 cm long on MS1 and MS3 media was 27.6 and 7.2%, respectively, while the shoot regeneration from transverse nodal stem segments was observed only in single explants. Similar results were achieved for leaf petioles when they were cultured on MS1 medium. Thus, on average, for all variants of culture media, the explants were arranged in the following sequence according to the frequency of in vitro somatic shoot organogenesis: 7-day-old hypocotyl fragments (52.5%) > 42-day-old nodal segments of stem (11.6%) > 42-day transverse nodal segments of stem (0.4%), leaf petioles (0.2%), and 21-day-old hypocotyl fragments (0%). 

Juvenile hypocotyl fragments were characterized not only by the maximum somatic shoot organogenesis frequency but by the highest number of regenerated shoots per explant (Figure 3b and Figure 4). Thus, when they were cultured on the MS2 and MS3 media, multiple shoot organogenesis was observed (5.7 and 5.6, respectively). The average number of regenerants for other explants varied between 1.0 and 1.5. In average values for all culture media variants, explants were arranged in the following sequence according to the number of regenerated shoots per explant: 7-day-old hypocotyl fragments (4.3) > 42-day-old nodal segments of stem (1.5) > 42-day transverse nodal segments of stem, leaf petioles (1.0). 

### 3.4. In Vitro Rooting of Regenerants and Plantlets Adaptation to Ex Vitro Conditions

Both variants of RIM were successfully applied for rooting of shoots regenerated from juvenile hypocotyl-derived callus tissue. The frequency of rhizogenesis was significantly higher during regenerant culture on ½ MS medium without PGRs (81%) compared to the RIM supplemented with 1 mg/L IBA (65.2%). Plantlets with a well-developed root system were acclimatized to ex vitro conditions. The efficiency of plantlet adaptation to soil conditions was 90% (Appendix A).

## 4. Discussion

To date, numerous highly efficient protocols for direct in vitro shoot regeneration of castor bean have been developed [19,20,25,28,32]. However, they are most suitable not for genetic transformation but for clonal micropropagation since they involve various tissues and organs with an apical or lateral shoot meristem as the initial explant source, for example, shoot tips [25,32], juvenile hypocotyl fragments with shoot apical meristem [19,20,25], embryo axes [32], and mature embryo [28]. Induction of in vitro callus-derived somatic shoot organogenesis from explants without an apical meristem has not yet become a routine technique due to the relatively low morphogenetic capacity. Some researchers have also reported an even lower frequency of callus formation [21,22]. Thus, cotyledons, epicotyls, immature leaves, and embryos of castor bean GC-3 and JP-65 lines have shown very low callus induction ability, while only the hypocotyl fragments were found as a responsive explant (on average for all studied culture media, the callus formation frequency for GC-3 and JP-65 genotypes was 47.1 and 45.1%, respectively) [21]. Additionally, the frequencies of organogenic callus formation from cotyledonary explants of *R*. *communis* L. cv. TMV 6, cultured on MS media supplemented with different cytokinins (1–3 mg/L kinetin and 6-(γ,γ,-dimethylallylamino)-purine (2iP), as well as 0.5–4.5 mg/L 6-BAP), varying from 5.2 to 25.3% [22]. On the contrary, the high callus formation ability (60–100%) from hypocotyls and cotyledon petioles of castor bean cvs. Zanzibar Green and Impala Bronzovaya were also established [23]. The current study demonstrated high responsiveness to callus formation of hypocotyl explants (≥99.0%) derived from seedlings of two different ages (7- and 21-day-old), 42-day-old nodal stem segments varying in size and explant preparation (≥92.4%), and leaf petioles (≥79.7%) (Figure 2). 

The morphogenetic responses of castor bean are influenced by exogenous PGRs and their interactions with endogenous phytohormones. In vitro shoot regeneration via both direct and indirect organogenesis was achieved on culture medium containing different types and concentrations of cytokinins such as TDZ [20,22,26], zeatin [18,23], 6-BAP [25], kinetin [25], and 2iP [27] alone or in combination with a low concentration of auxin (IAA [23,25] or 1-naphthylacetic acid (NAA) [22,25]). Some authors propose to simultaneously combine two cytokinins in the shoot induction medium for improving regeneration capacity, for example, BAP and kinetin [21,24,33] or BAP and TDZ [18]. The application of 15 mg/L glutamine [22] and gibberellic acid [19,21,22] was also recommended as additional culture medium components for the formation and elongation of regenerants. Additionally, we previously found that including the ethylene inhibitor AgNO_3_ at a concentration of 5 mg/L has a beneficial effect on indirect shoot organogenesis (unpublished data). According to our previous studies, we used MS basal culture medium supplemented with 0.1 mg/L IAA and 5 mg/L AgNO_3_, and 1 mg/zeatin (MS1), 0.25 mg/L TDZ (MS2), and 5 mg/L 6-BAP (MS3) as cytokinin source to induce indirect somatic shoot organogenesis from various explants. Consequently, it was estimated that the highest regenerative capacity is characterized by juvenile hypocotyl fragments cultured on MS2 medium. This treatment produced multiple shoot organogenesis (5.7) with a frequency of 67.5% (Figure 3a,b). Additionally, radically age-related differences in the efficiency of in vitro shoot regeneration ability from hypocotyl-derived callus have been convincingly proven. At the same time, the shoot organogenesis frequency from leaf petioles of the castor bean cv. Zanzibar Green (27.6%) was comparable to the value of cotyledon petioles (16.5%) [23]. 

An analysis of the literature indicates the availability of radically different experimental data on the in vitro root formation capacity of castor bean regenerants. Some authors indicate a high frequency of rhizogenesis (more than 85%) during rooting on agar-solidified ½ MS culture medium with varying concentrations of auxin, for example, 0.2 mg/L IAA [19], 1 mg/L IBA [18], and 5 mg/L IBA or NAA [20]. In the case of regenerants obtained from the epicotyl explants of castor bean cv. VERC-02, root formation with a frequency of 36.7% was established on the control culture medium without PGRs [18]. In contrast, Zalavadiya et al. (2014) did not achieve rooting of shoots on RIM containing IAA, IBA, and AgNO_3_ singly or in combinations even after 30 days of culture [21]. As an alternative, an in vitro grafting procedure has been developed for the rooting of regenerants and subsequent efficient adaptation of plantlets to soil conditions [33]. In the current study, we successfully induced rhizogenesis in shoots regenerated from juvenile hypocotyl-derived callus tissue on the RIM, both without PGRs and supplemented with 1 mg/L IBA. At the same time, the frequency of rhizogenesis was significantly higher during 30 days of culture on RIM without PGRs (81%) compared to ½ MS basal medium containing exogenous auxin (65.2%). Castor bean plantlets with well-developed root systems were efficiently acclimatized to soil conditions with a frequency of 90% (Appendix A). 

Thus, an effective and rapid protocol for in vitro indirect shoot organogenesis has been developed using hypocotyl fragments derived from 7-day-old aseptic donor seedlings of the castor bean cv. Zanzibar Green (Figure 5). 

The utilization of juvenile seedlings obtained from surface-sterilized mature seeds at an early germination stage significantly reduces the time period required to produce donor explants. It was found that 7- and 21-day-old hypocotyl fragments, as well as two types of nodal stem segments obtained from adult aseptic seedlings, are characterized by a high explant viability and callus formation capacity with a frequency of 79.7–100% (Figure 2). However, the success of in vitro somatic shoot organogenesis was significantly determined not only by the culture medium composition and type of explant but also depending on its age, as well as on the size and explant preparation in cases of hypocotyl and age-matched nodal stem fragments, respectively (Figure 3a). Multiple somatic shoot organogenesis (5.7 regenerants per explant) with a frequency of 67.5% was achieved through three subcultures of callus tissue derived from 7-day-old hypocotyl fragments on MS basal medium containing 0.25 mg/L TDZ, 0.1 mg/L IAA, and 5 mg/L AgNO_3_. Castor bean regenerants were successfully rooted on ½ MS basal medium without PGRs (81%). In vitro plantlets with well-developed roots were adapted to ex vitro conditions with a frequency of 90%. 

## 5. Conclusions

An effective and rapid protocol for in vitro indirect shoot organogenesis has been developed using hypocotyl explants derived from 7-day-old aseptic donor seedlings of the castor bean cv. Zanzibar Green. Multiple somatic shoot organogenesis (5.7 regenerants per explant) with a frequency of 67.5% was achieved through three subcultures of juvenile hypocotyl-derived callus on MS basal medium containing 0.25 mg/L TDZ, 0.1 mg/L IAA, and 5 mg/L AgNO_3_. This protocol can be successfully applied to induce an indirect shoot formation in different *R. communis* L. genotypes, as well as for genetic transformation or CRISPR/Cas-based genome-edited castor bean plants with improved valuable traits. 

## Figures and Tables

**Figure 1 biotech-13-00025-f001:**
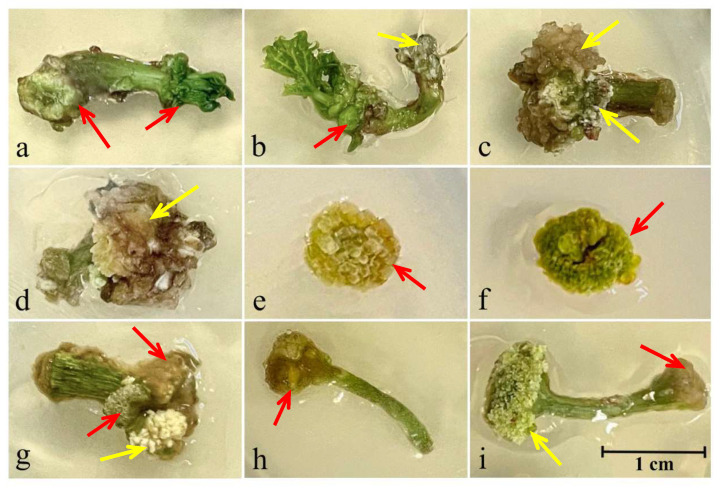
Morphological characteristics of callus tissue formed on 7-day-old (**a**,**b**) and 21-day-old (**c**,**d**) hypocotyl fragments, 42-day-old transverse nodal segments of stem (**e**,**f**), 42-day-old nodal segments of stem (**g**), and petioles of first true leaves (**h**,**i**) of castor bean cv. Zanzibar Green cultured on the MS1 (**d**,**e**,**g**), MS2 (**a**,**f**,**i**), and MS3 (**b**,**c**,**h**) media. Red and yellow arrows indicate dense and friable calli, respectively.

**Figure 2 biotech-13-00025-f002:**
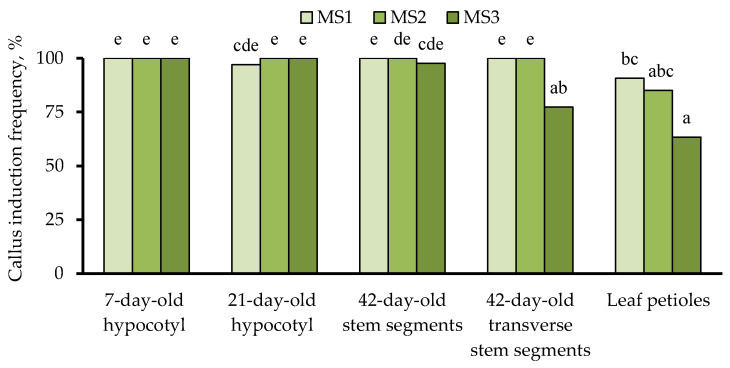
Effects of culture medium composition and explant source of castor bean cv. Zanzibar Green on the callus formation frequency. Treatments followed by at least one letter do not differ significantly at α = 0.05, according to Duncan’s multiple range test. MS1–MS3—Murashige and Skoog basal medium supplemented with 0.1 mg/L IAA, 5 mg/L AgNO_3_, and 1 mg/L zeatin (MS1), 0.25 mg/L TDZ (MS2), and 5 mg/L 6-BAP (MS3).

**Figure 3 biotech-13-00025-f003:**
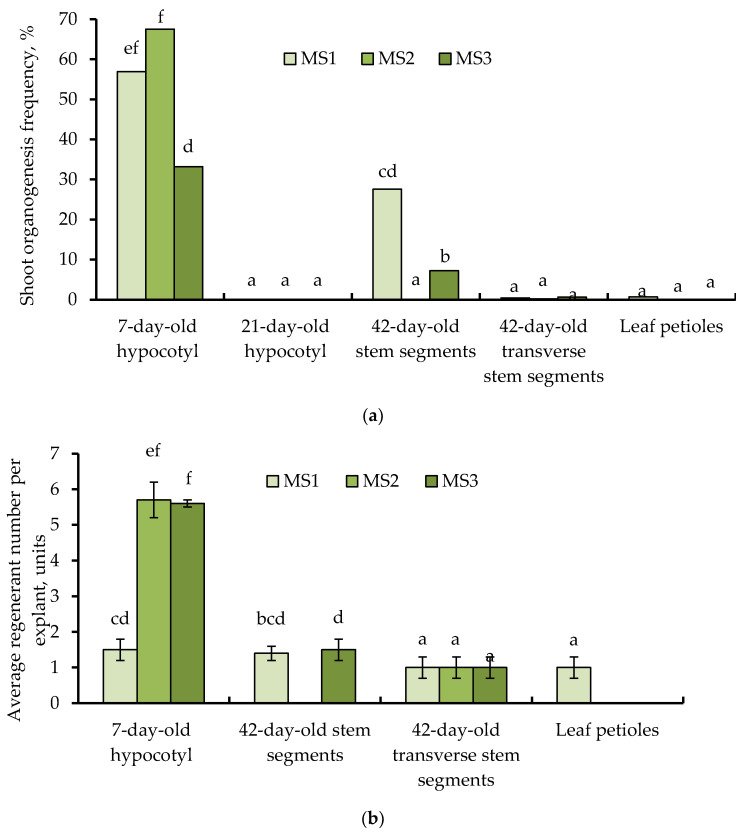
Effects of culture medium composition and explant source of castor bean cv. Zanzibar Green on in vitro shoot organogenesis frequency (**a**) and average number regenerants per explant (**b**). Treatments followed by at least one letter do not differ significantly at α = 0.05, according to Duncan’s multiple range test. MS1–MS3—Murashige and Skoog basal medium supplemented with 0.1 mg/L IAA, 5 mg/L AgNO_3_, and 1 mg/L zeatin (MS1), 0.25 mg/L TDZ (MS2), and 5 mg/L 6-BAP (MS3).

**Figure 4 biotech-13-00025-f004:**
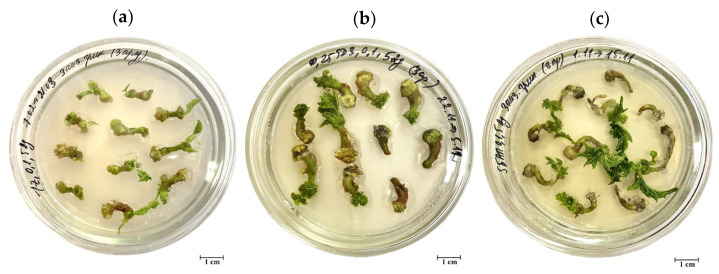
Induction of somatic shoot organogenesis from juvenile hypocotyl-derived callus of castor bean cv. Zanzibar Green on MS1 (**a**), MS2 (**b**), and MS3 (**c**) culture media.

**Figure 5 biotech-13-00025-f005:**
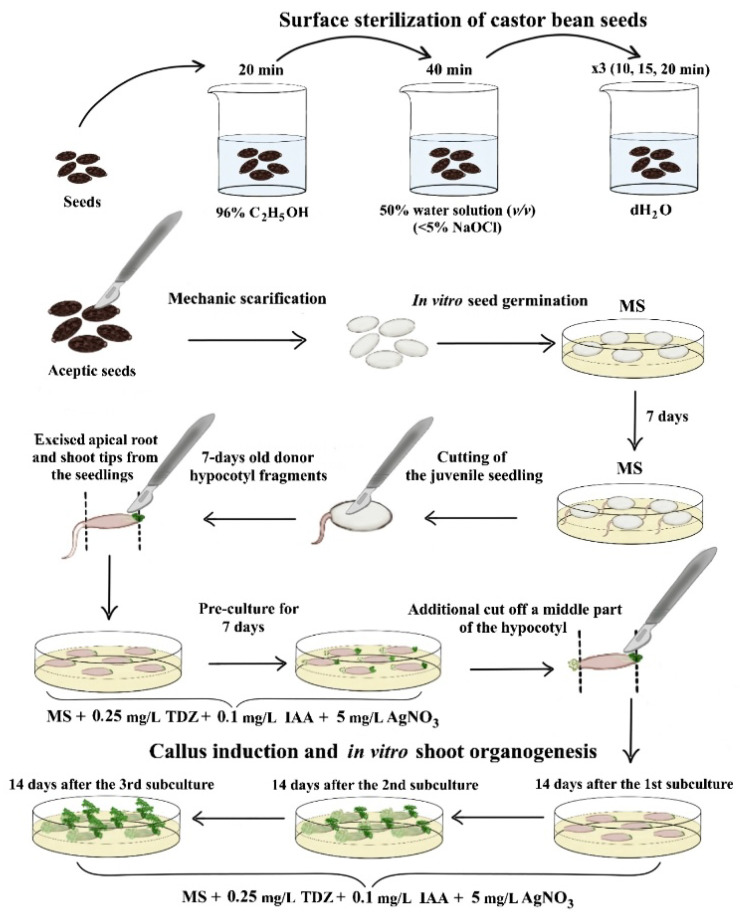
Scheme of induced indirect in vitro shoot organogenesis using donor juvenile hypocotyl explants derived from aseptic seeds of the castor bean cv. Zanzibar Green.

**Table 1 biotech-13-00025-t001:** Influence of culture medium composition on the explant viability of castor bean cv. Zanzibar Green.

Explant Type	Number of Explants (Units)		Explant Viability (%)
	Total	Necrotic	
MS1 (1 mg/L zeatin + 0.1 mg/L IAA + 5 mg/L AgNO_3_)			
7-day-old hypocotyl fragments	51	0	100 e
21-day-old hypocotyl fragments	216	18	91.7 cde
42-day-old nodal segments of stem	54	0	100 e
42-day-old transverse nodal segments of stem	162	0	100 e
Petioles of first true leaves	288	53	81.6 bc
MS2 (0.25 mg/L TDZ + 0.1 mg/L IAA + 5 mg/L AgNO_3_)			
7-day-old hypocotyl fragments	66	0	100 e
21-day-old hypocotyl fragments	78	0	100 e
42-day-old nodal segments of stem	30	0	100 e
42-day-old transverse nodal segments of stem	144	0	100 e
Petioles of first true leaves	78	13	83.3 abc
MS3 (5 mg/L 6-BAP + 0.1 mg/L IAA + 5 mg/L AgNO_3_)			
7-day-old hypocotyl fragments	90	0	100 e
21-day-old hypocotyl fragments	30	0	100 e
42-day-old nodal segments of stem	90	6	93.3 cde
42-day-old transverse nodal segments of stem	60	15	75.0 ab
Petioles of first true leaves	30	11	63.3 a

Treatments followed by at least one letter do not differ significantly at α = 0.05, according to Duncan’s multiple range test.

**Table 2 biotech-13-00025-t002:** Morphological characteristics of callus tissue depending on the explant type and culture medium composition.

Explant Type	Morphological Characteristics of Callus Tissue			Morphogenetic Response ^c^
	Color ^a^	Consistency ^b^	Site of Callus Formation	
MS1 (1 mg/L zeatin + 0.1 mg/L IAA + 5 mg/L AgNO_3_)				
7-day-old hypocotyl fragments	G	D	Cut from the shoot meristem side	+
	B, W	F	Cut from the root meristem side	–
21-day-old hypocotyl fragments	Y-B	F	At both ends of the cut	–
42-day-old nodal segments of stem	Y	D	At both ends of the cut	+
42-day-old transverse nodal segments of stem	Y-B	D	Whole explant	+
Petioles of first true leaves	LB	D	At both ends of the cut	+
MS2 (0.25 mg/L TDZ + 0.1 mg/L IAA + 5 mg/L AgNO_3_)				
7-day-old hypocotyl fragments	G	D	Cut from the shoot meristem side	+
	LG	D	Cut from the root meristem side	–
21-day-old hypocotyl fragments	B, W	F	At both ends of the cut	–
42-day-old nodal segments of stem	Y-B	D	At both ends of the cut	–
42-day-old transverse nodal segments of stem	G	D	Whole explant	+
Petioles of first true leaves	B	D	Cut from the leave blade side	–
	LG	F	Cut from the stem side	–
MS3 (5 mg/L 6-BAP + 0.1 mg/L IAA + 5 mg/L AgNO_3_)				
7-day-old hypocotyl fragments	G	D	Cut from the shoot meristem side	+
	B, W	F	Cut from the root meristem side	–
21-day-old hypocotyl fragments	B, W	F	At both ends of the cut	–
42-day-old nodal segments of stem	G, B	D	At both ends of the cut	+
42-day-old transverse nodal segments of stem	Y-B	D	Whole explant	+
Petioles of first true leaves	LB	D	Cut from the stem side	–

Notes. ^a^: W—white; Y—yellow; Y-B—yellow-brown; G—green; B—brown; LB—light brown; LG—light green. ^b^: D—dense; F—friable. ^c^: «+» and «–»—morphogenic and non-morphogenic calli, respectively.

**Table 3 biotech-13-00025-t003:** Two-way ANOVA test to evaluate the significance of culture medium components and explant type on the callus formation frequency.

Source of Variation	ss	df	ms	F_05_	F
Total	9251.9	44	–	–	–
Variants	6489.3	14	463.5	2.01	5.03 *
Factors					
A (culture medium)	972.8	2	486.4	3.32	5.28 *
B (explant)	3914.1	4	978.5	2.69	10.63 *
Interaction AB	1602.4	8	200.3	2.27	2.18
Error	2762.6	30	92.1	–	–

Notes: ss—sum of squares, df—degrees of freedom, ms—mean square, F_05_—critical F value at 5% significance level (α = 0.05), F—F value, *—F test significant at α = 0.05 (F > F_05_). The percentage values of callus induction frequency were arcsin $\;\sqrt{X}$ transformed prior to the ANOVA test.

## Data Availability

The experimental data obtained and analyzed during estimation are included in this article.

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
