# Peer review of "An Efficient and Rapid Protocol for Somatic Shoot Organogenesis from Juvenile Hypocotyl-Derived Callus of Castor Bean cv. Zanzibar Green"

_biotech, 2024, doi:10.3390/biotech13030025_

Round 1
Reviewer 1 Report
Comments and Suggestions for Authors
The results obtained are positive in terms of shoot regeneration, but not very relevant, as it is known than young tissues respond better than older one, overall respect hypocotyls or cotyledons. On the other hand, there are already many references obtainig regeneration of shoots with cytokin plus an auxin in this plant species.
Author Response
Dear Reviewer,
On behalf of ourselves and the co-authors, we are grateful for the time you spent on our manuscript, as well as your high assessment of our work. In addition, we also send a resubmitted Word document with your and other reviewers' comments. We hope that the resubmitted version of the article has become better and more understandable to readers.
Best regards,
Marat Khaliluev
Reviewer 2 Report
Comments and Suggestions for Authors
Dear authors,
I think this is a very good work supported by the large data scale for all tested parameters. I miss some pictures of the protocol, and perhaps of the calli wanted in the research. All other details are well described.
I think the used literature in this article is relevant and in the scope of the production of shoot organogenesis.
Statistics is well performed only put the year of the program and pput the refenrence in the reference list.
In the sense of genetic transformations, this is a big effort to find a efficient protocol. I hope it will be used in the future...
I highly recommend to publish this type of work,
Regards
Zvjezdana
Author Response
Dear Zvjezdana,
On behalf of ourselves and the co-authors, we are grateful for the time you spent on our manuscript, as well as your high assessment of our work. We thank the reviewer for the minor valuable comments that we made to the resubmitted manuscript. We indicated the year the statistical program was created (L 179).
Once again, we are so grateful for your review and valuable comments. In addition, we also send a resubmitted Word document with your and other reviewers' comments. We hope that the resubmitted version of the article has become better and more understandable to readers.
Best regards,
Marat Khaliluev
Reviewer 3 Report
Comments and Suggestions for Authors
Dear authors,
The manuscript has to be improved before considering it for publication. Please, go through the manuscript and address the following point.
Line 31 – 33: Arrange the Keywords in alphabetical order:
Ricinus communis L., explant source, plant growth regulators (PGRs), morphological characteristics of callus tissue, efficiency of indirect in vitro shoot organogenesis, shoot rhizogenesis, plantlets adaptation to ex vitro conditions.
Line 195 – 196: All (explant viability) percentage values after the whole number were separated by a comma; instead, a dot must be used throughout the manuscript (Table 1).
Line 261 – 265: In the caption of the figure, please spell out what the MS1, MS2, and MS3 are referring to (Murashige and Skoog?). Follow the same in the rest of the figures (2 and 3).
Line 191 – 193: “Regardless of the cytokinin component used in the culture medium to induce the morphogenetic responses, the viability of leaf petioles was significantly lower (63.3–83.3%) compared to other explants.” This statement is NOT clear; it needs to be paraphrased to which treatment is referring. If not from “culture medium”, does the effect come from the other (explant” factor?
Line 385 – 386: I am not clear whether Figure 5 is a methodology (procedure) or a finding. If it is a methodology, please move this schematic illustration to section 2.3. of the materials and methods of the manuscript.
Line 406 – 410: Your findings are more than the conclusion. There are a lot of messages that need to be conveyed to prospective readers. For example, you can add which treatment is better for inducing an indirect shoot formation in different genotypes of the current study. You can also add which explant can produce better transgenic lines or which explant can be better for CRISPR/Cas-based genome editing in producing valuable traits of the plant of the current plant material.
With regards,
the reviewer

Author Response
Dear Reviewer,
On behalf of ourselves and the co-authors, we thank you for your appreciation of our manuscript and valuable comments. We thank the reviewer for his high assessment of our manuscript. We are confident that your comments and corrections will make our manuscript better.
Remark 1: Line 31 – 33: Arrange the Keywords in alphabetical order:
Ricinus communis L., explant source, plant growth regulators (PGRs), morphological characteristics of callus tissue, efficiency of indirect in vitro shoot organogenesis, shoot rhizogenesis, plantlets adaptation to ex vitro conditions.
Response 1: Thank you for your valuable comment. We agree with the reviewer's remark. We have arranged the Keywords in alphabetical order (L 31-33):
Keywords: efficiency of indirect in vitro shoot organogenesis, explant source, morphological characteristics of callus tissue, plant growth regulators (PGRs), plantlets adaptation to ex vitro conditions, Ricinus communis L., shoot rhizogenesis.
Remark 2: Line 195 – 196: All (explant viability) percentage values after the whole number were separated by a comma; instead, a dot must be used throughout the manuscript (Table 1).
Response 2: We thank the reviewer for valuable comment. We have replaced the comma in the numeric values in Table 1 (L. 195).
Remark 3: Line 261 – 265: In the caption of the figure, please spell out what the MS1, MS2, and MS3 are referring to (Murashige and Skoog?). Follow the same in the rest of the figures (2 and 3).
Response 3: Thank you for your valuable comment. We agree with the reviewer's remark. We have added a caption to the Figures 2 (L 265-267) and 3 (L 289-291).
MS1–MS3 – Murashige and Skoog basal medium supplemented with 0.1 mg/L IAA, 5 mg/L AgNO3, as well as 1 mg/L zeatin (MS1), 0.25 mg/L TDZ (MS2) and 5 mg/L 6-BAP (MS3).
Remark 4: Line 191 – 193: “Regardless of the cytokinin component used in the culture medium to induce the morphogenetic responses, the viability of leaf petioles was significantly lower (63.3–83.3%) compared to other explants.” This statement is NOT clear; it needs to be paraphrased to which treatment is referring. If not from “culture medium”, does the effect come from the other (explant” factor?
Response 4: Thank you for your valuable comment. We've paraphrased for better understanding (L 191-192).
The viability of leaf petioles was significantly lower (63.3–83.3%) compared to other explants when cultured
on MS1–MS3 induction media.
Remark 5: Line 385 – 386: I am not clear whether Figure 5 is a methodology (procedure) or a finding. If it is a methodology, please move this schematic illustration to section 2.3. of the materials and methods of the manuscript.
Response 5: Indeed, Fig. 5 is a finding. This scheme is the resultant one. It summarizes all the stages of indirect in vitro shoot organogenesis using donor juvenile hypocotyl explants, derived from aseptic seeds of the castor bean cv. Zanzibar Green. That's why we included it in the Discussion section.
Remark 6: Line 406 – 410: Your findings are more than the conclusion. There are a lot of messages that need to be conveyed to prospective readers. For example, you can add which treatment is better for inducing an indirect shoot formation in different genotypes of the current study. You can also add which explant can produce better transgenic lines or which explant can be better for CRISPR/Cas-based genome editing in producing valuable traits of the plant of the current plant material.
Response 6: Thank you for your valuable comment. We have detailed and expanded the Conclusion section according to your recommendations (L. 412-420).
An effective and rapid protocol for in vitro indirect shoot organogenesis has been developed using hypocotyl explants derived from 7-day-old aseptic donor seedlings of the castor bean cv. Zanzibar Green. Multiple somatic shoot organogenesis (5.7 regenerants per explant) with a frequency of 67.5% was achieved through 3 subcultures of juvenile hypocotyl-derived callus on MS basal medium containing 0.25 mg/L TDZ, 0.1 mg/L IAA and 5 mg/L AgNO3. This protocol can be successfully applied to induce of an indirect shoot formation in different R. communis L. genotypes, as well as to produce transgenic lines or CRISPR/Cas-based genome editing in castor bean plants with improved valuable traits.
Once again, we are so grateful for your review and valuable comments. In addition, we also send a resubmitted Word document with your and other reviewers' comments. We hope that the resubmitted version of the article has become better and more understandable to readers.
Best regards,
Marat Khaliluev.

Round 2
Reviewer 1 Report
Comments and Suggestions for Authors I have looked through the revised version and I have seen only a few minor changes. I am sorry, but perhaps you would have to send the article to a third reviewer.I reassert on my first decision.
Author Response
We thank the reviewer for reading the article and rating it.
Best regards,
Marat Khaliluev